

# Experimental determination of the flood wave transformation and the sediment resuspension in a small regulated stream in an agricultural catchment

David Zumr[1], Tomáš Dostál[1], Jan Devátý[1], Petr Valenta[1], Pavel Rosendorf[2], Alexander Eder[3], Peter Strauss[3]

[1]Faculty of Civil Engineering, Czech Technical University in Prague, Prague 6, 16629, Czech Republic
[2]T. G. Masaryk Water Research Institute, Prague, 16000, Czech Republic
[3]Institute for Land & Water Management Research, Federal Agency for Water Management, 3252 Petzenkirchen, Austria

*Correspondence to*: David Zumr (david.zumr@fsv.cvut.cz)

**Abstract.** This paper presents the methodology used for artificial flood experiments conducted in a small artificial, trained (regulated) channel on the Nučice experimental agricultural catchment (0.5 km$^2$), central Czech Republic, and the results of the experiments. Two series of experiments were carried out in contrasting initial conditions: (a) in summer, when the stream banks were dry, the baseflow was negligible and the channel was fully overgrown with vegetation; and (b) in spring, when the stream banks were almost water saturated, the baseflow was above the annual average, and there was no vegetation present. Within each campaign, three successive flood waves, each with an approximate volume of 17 m$^3$ and peak flow of ca 40 l s$^{-1}$, were pumped into the upper part of the catchment drainage channel. The transformation of the flood wave and the sediment transport regime within an approximately 400 m long channel section were monitored by measuring the discharge, the turbidity and the electrical conductivity in three profiles along the stream. On the basis of the results, it was concluded that there is a considerable amount of deposited sediment in the channel that can be re-mobilized even by small floods. Part of the recorded sediment therefore originates from the particles deposited during previous soil erosion events. The flood waves initiated in dissimilar instream conditions progressed differently – we show that the saturation of the channel banks, the stream vegetation and the actual baseflow had a strong influence on the flood transformation and the sediment regime in the channel.

## 1 Introduction

Excessive soil erosion from upland areas resulting in the transport of soil particles with bound organic matter, nutrients, microbes or pollutants into the rivers and reservoirs is considered as a major environmental problem (Drummond et al., 2014; Lal et al., 2007; Pimentel et al., 1995; Stoate et al., 2001). The processes of soil particle mobilization and transportation within agriculturally-used fields, including the transfer into streams and rivers, have been extensively studied (Báčová and Krása, 2016; Boardman, 2003; Lal, 1998; Neal and Anders, 2015).

The headwater streams and drainage channels in sediment source areas, typically small rural catchments with intensively cultivated soils, have considerable retention capacities for sediment and nutrients (Hession et al., 2003). The streams are narrow





with a small flow profile, the baseflow is usually low, the stream bed contains fine-grained particles with a high concentration of nutrients, and extensive vegetation can therefore often be found there. Soil particles that enter the channel during an erosion event can easily get stacked and be remobilized during subsequent runoff events, and the nutrients tend to be retained (Withers and Jarvie, 2008). The source of the sediment and the processes related to the suspended sediment dynamics in the closing

profiles are therefore of fundamental importance for an assessment of the sediment budget and the transport of dissolved or absorbed substances in the catchment (Walling, 2005). However, even physically-based mathematical models of soil erosion assume that the sediment transported through water courses originates from a recent (or current) rainfall-runoff event. Similarly, traditional experiments and soil erosion monitoring usually rely on measurements of the sediment yield at the catchment outlet, assuming that the measured sediment yield originates on the hillslopes. If any retention in the channel is

expected, no resuspension is then assumed, and this affects the total sediment budget. (Minella et al., 2008) point out that the transport capacity of the channel may increase, and that the stream bed sediment is easily mobilized during runoff events with no eroded sediment from the catchment. (Zumr et al., 2015) and also (Musolff et al., 2015) show that a quick runoff response with no soil erosion on the fields is very commonly observed on cultivated catchments where subsurface runoff or tile drains are the dominant controls. The resuspension regime of the stream bed sediment and the connected nutrient transport depend

on the characteristics of the stream, the hydrograph of the flood wave and the actual conditions of the channel (Peterson and Benning, 2013).

The sources of the suspended sediments recorded at the catchment outlet also vary in course of the season due to seasonally varying vegetation (Hearne et al., 1994). The development of aquatic macrophytes limits the discharge capacity of the channels. (Keesstra et al., 2012) evaluated the effect of temporary variable vegetation cover within the natural and semi-natural

headwater channels and the stream riparian zone on water and sediment transport. On the basis of numerical modelling, they concluded that vegetation affects resuspension especially during high flow conditions in streams that are not sediment supply limited. Similarly (Huisman et al., 2013) showed that the sediment resuspension within headwater channels is more important during the later parts of the year, when the vegetation is dense. (Shore et al., 2015) showed that in the case of well-trained channels there is greater potential for fast sediment transportation downstream. However, this is not necessarily the rule in

sparsely maintained and over-vegetated channels, where the sediment retention capacity is not negligible.

The key questions that we will address here are:

• Can well-trained and well-regulated stream channels act as a temporal sediment trap and sediment source due to the resuspension of sediments deposited from previous erosion events?

• How does the flood wave transformation regime and the suspended solids remobilization regime change within one season as a consequence of various instream vegetation and baseflow conditions?

• How does the resuspended sediment concentration and the mass movement change in the event of repeated short flood waves?



To answer these questions, we initiated two sets of three small artificial floods into a typical drainage channel in the rural landscape of central Bohemia, Czech Republic. The experiments were performed recurrently in summer and in winter, when the channel vegetation, the baseflow and the channel saturation differ most.

## 2 Methods

### 2.1 Site description

The experiments were performed in the stream which drains the Nučice rural experimental catchment, Czech Republic (Fig. 1). The Nučice catchment (49° 57' 49.230" N, 14° 52' 13.242" E) was established in 2011 with the main purpose to monitor and study the rainfall-runoff and water soil erosion processes originating from intensive rainfall over cultivated fields (Zumr et al., 2015).

The catchment 0.531 km2 in size is at elevations ranging from 382 m to 417 m a.s.l. The inclination of the slopes varies between 1 % and 12 %, the average slope being 3.9 %. The annual average precipitation is 630 mm, and the annual evapotranspiration is 500 mm. The mean air temperature is 6 °C, and the climate is considered as humid continental. The catchment is unique with its very uniform land use. More than 95 % of the area is arable land, while the remaining parts are the watercourse, riparian trees and shrubs and paved roads. There are no forests, grassland or urbanized areas. The arable land is cultivated down to the stream banks, and conservation tillage is practised. The usual crops are winter wheat (Triticum aestivum), mustard (Sinapis alba L) and rapeseed (Brassica napus). The soils are classified as Cambisols and Luvisols. The topsoil has a loamy texture with a mean of 13 % clay, 42 % silt and 45 % sand. The average annual topsoil saturated hydraulic conductivity is 4.8 $10^{-7}$ m s$^{-1}$, and the mean organic carbon content is 1.9 %.

The Nučice catchment is drained by an artificially trained narrow stream, which has been piped in the uppermost part. The channel was modified into its current form in the 1950s, with the aim to decrease the groundwater level and to prevent inundation of the fields. The piped section is 530 m in length, and the open channel down to the outlet profile of the experimental catchment extends to 424 m. The straight, deep channel is in direct contact with the surrounding fields. The riparian vegetation is only sparse.

The channel has a trapezoid profile which is 0.6 m in width at the stream bed, and the slope of the banks is 1:2. The stream bed and footslopes up to 0.3 m are stabilized with concrete tiles. There are two culverts on the stream. One is 56 m from the start of the open channel, and it is 0.8 m in inner diameter and 10.2 m in length. The second culvert is 337 m from the start, and is 0.6 m in diameter and 7.8 m in length. The average depth of the channel is 1.5 m. Thecurrent situation of the channel represents very well the situation in most small regulated drainage channels in the country: there has been very little maintenance during the last ca 30 years. Locally, therefore, the stabilization is defective and the channel profile has been covered by extensive weed vegetation with a predominance of stinging nettles (Urtica dioica), orchard grass (Dactylis glomerata), pigweed (Chenopodium album) and hogweed (Heracleum sphondylium).



The typical flow conditions at the gauging station, as observed during the period of monitoring (2011 to 2016), are as low as 0 1 s$^{-1}$ to 0.2 1 s$^{-1}$ during the summer months and around 4 1 s$^{-1}$ towards the end of winter. Summer storm-runoff events accompanied by an increase in the concentration of suspended solids in the runoff are characteristic events. The storms cause a short, steep wave with a concentration time of 50 to 240 minutes (Zumr et al., 2015).

## 2.2 Experimental setup

The experiments were performed in the open section of the channel (Fig. 1). The total monitored length was 424 m, which is the distance between the injection profile (profile S) and the basin closing profile (profile C). The point where we injected water is considered as the start (0 m), and the basin outlet is considered as the end (424 m). The release profile was placed directly at the beginning of open channel section. Water was pumped into the stream over a period of 7 minutes simultaneously from a filled water reservoir and from the onboard supply of a fire truck, using four fire hoses. We used drinking water from a nearby reservoir. The total pumped water volume for each wave was approximately 17 m3. The very first wave (W1) was initiated with ca 15 m$^3$ of water, because the pump could not draft the remaining 2 m$^3$ from the bottom of the water reservoir. The discharge fluctuated around 40 1 s$^{-1}$, and no significant peaks were produced during injection. The water was pumped first into a small stilling basin to prevent excessive disturbance to the streambed, and to be able to make precise measurements of the discharge produced by the pump. The discharge was measured with a mobile H flume, where the water level was recorded automatically with a pressure transducer.

We established three monitoring stations along the watercourse (Figs. 1 and 2) to monitor the discharge and the electrical conductivity, and to collect samples for measuring the concentration of the suspended solids. After taking the initial water sample for an evaluation of the baseflow water properties, we started the water sampling in each experimental profile immediately after the flood wave arrived. Samples approximately 1 l in volume were taken every minute during the rise, the peak and shortly after the decrease of the discharge. After that, the sampling interval was reduced to 2 to 5 minutes intervals to obtain approximately 30 samples for each wave and observation profile. The samples were analysed in the laboratory. The concentration of suspended solids, phosphorus and nitrogen were measured. Each station was equipped with a pre-programmed camera on a tripod to confirm the exact time of arrival of the wave and to document the progress of the wave and the sampling.

The first 66 m long section between pumping station S and profile A was meant to be used for wave dispersion and fluctuation stilling due to non-homogeneities in the pumping process. Station A was situated at the upper outlet of the culvert. The culvert ends with a free outflow, where the discharge was measured both volumetrically and hydrometrically. Monitoring profile B, in a distance of 224 m, was equipped with a rectangular weir, and the discharge was estimated on the basis of the measured depth of the water and the known rating curve. Profile C was located at the outlet of the experimental basin, positioned 424 m from the beginning of open channel section. The outlet is permanently equipped with an H flume with capacity of up to 400 l s$^{-1}$. The water depth was measured simultaneously with a pressure transducer and with an ultrasonic probe every ten seconds. The sediment concentration was measured in the lab on samples taken from the outlet and with a turbidity probe (ViSolid 700IQ, WTW, Germany) installed in a stilling basin below the H flume. The onsite data were logged automatically with a





CR1000 datalogger (Campbell Sci., UK). The electrical conductivity of the water in all profiles was measured with HQ40D portable multimeters (Hach Lange, Germany).

The experiments were conducted in September 2012 and in March 2013. Within each of the campaigns we carried out three wave experiments (W1 to W3 in September, W4 to W6 in March). Subsequent waves were always initiated after the discharge in the outlet profile (station C) had dropped close to the initial baseflow. The second waves in the series (W2 and W5) were enriched with NaCl as a tracer to compare the water velocity and the water celerity during wave propagation. The tracer was dissolved in the water reservoir to obtain a concentration of approximately 6 g l$^{-1}$. The first and third waves in each set contained no tracer.

The general conditions within the catchment and the stream prior to the experiments differed in September 2012 and in March 2013 (Table 1). In September, the stream baseflow was at its annual minimum, the soil water content was below its field capacity, and the instream vegetation was densely overgrown. In March 2013, the baseflow was at its annual maximum because of saturated soil from the snow melting, the instream vegetation was sparse, and the remaining plants were flattened on the stream bed and banks (Fig. 3).

## 3 Results

The hydrographs and sedigraphs of all six waves are shown in figs. 4-6. The shape characteristics of the waves and the transformation are summarized in Table 2. The water and sediment balance are presented in Table 3.

### 3.1 Water flow regime

The hydrographs of the subsequent experiments differed on all the monitoring profiles. The velocity of the waves and the maximum flow rates increased between the successive waves (fig.4). All the waves approached the A profile less than 5 minutes after the start of the experiments, and the subsequent waves reached the profile slightly earlier. The time difference between the approaches of waves W1 and W3 is 41 s. Waves W1 and W2 reached a similar peak discharge of approximately 30 l s$^{-1}$, and wave W3 reached a peak discharge of 36.3 l s$^{-1}$. The difference was mainly caused by the slightly fluctuating rate of water pumping and by transient filling of the depression storages on the channel bed during the first wave experiment. Water exfiltration into the hyporheic zone contributed only a little over such a short distance and period of time, as was also observed by (Exner-Kittridge et al., 2016).

The time lags in the B profile already differed. Wave W1 arrived after almost 20 minutes. Waves W2 and W3 were faster, and appeared 15 minutes after pumping began. The peak discharge also increased, with subsequent waves starting at 12.8 l s$^{-1}$ for W1 and reaching 19.6 l s$^{-1}$ for W3. The difference between the arrival times of W1 and W3 in the C profile was also 5 minutes, and the maximum flow rate increased from 12.3 l s$^{-1}$ (W1) to 19.6 l s$^{-1}$ (W3). The volume of water that reached the closing profile was 9.8 m$^3$ for W1, 13.7 m$^3$ for W2 and 14.8 m$^3$ for W3. Within W1, only 69 % of the pumped water was recovered in the C profile. For W2 and W3, the recovery rate was 90 %.





The wave celerity along the stream was calculated according to the wave arrival time, which we defined as the time of the first rise of the hydrograph (Table 2). The average wave celerity for W1 was 0.20 m s$^{-1}$, for W2 it was 0.23 m s$^{-1}$, and for W3 the celerity was 0.24 m s$^{-1}$. The water flow velocity was calculated on the basis of the time of arrival of the tracer. The tracer was always detected later than the rise in the hydrograph (fig. 5). The mean water flow velocity was 0.15 m s$^{-1}$.

The hydrographs of waves W3-W6 are very similar to each other, and the time lags differ by less than one minute. The waves approached profile A after 3 minutes, profile B after 9 minutes, and profile C after 16 minutes. The peak discharge values observed in the individual profiles were also similar, but the last wave, W6, reached slightly higher values. The average peak discharge of W4 to W6 in profile A was 32.5 l s$^{-1}$, in profile B the peak discharge was 29.3 l s$^{-1}$, and in profile C the peak discharge was 26.5 l s$^{-1}$. 85 % of the initially pumped water volume was recovered in the C profile in all the March experiments,

which is similar to the results for waves W2 and W3 (in the summer experiment). The waves propagated with average celerity of 0.44 m s$^{-1}$, which is twice the velocity of the September waves.

## 3.2 Numerical modelling of the propagation of the waves

For a quantitative evaluation of the impact of the vegetation on the transformation of the flood wave, we built a simple 1D hydraulic model in the HEC-RAS code. The variable parameter between the September and March experiments was the stream

roughness factor (Manning n), which we attribute as a proxy of the actual character and density of the vegetation (Brookes, 1986). The approach is similar to (Nikora et al., 2008), who showed that the flow resistance is determined mainly by the general characteristics of the bulk instream vegetation, rather than by individual species. We assume that the vegetation resistance is caused mainly by the stem blockage factor, rather than by frictional energy losses, as the dense canopy occupies the cross section of the channel (Green, 2005).

The geometry of the channel was based on a set of 28 measured cross sections obtained by land surveying. To improve the precision of the simulations and to overcome numerical stability problems, intermediate profiles were added to the model by geometrical interpolation between the original cross sections. The average distance between cross sections in the final geometry set is 0.5 m. Moreover, each cross section was extended by a narrow Preismann bottom slot to deal with numerical stability issues in cases of very low discharges (propagation of the wave in an originally dry channel). The model was loaded

on the upper end with the measured flow. The rating curve based on the Chezy equation for uniform flow was used as a downstream boundary condition.

For each simulation, the simulated flow hydrograph at the downstream end of the model was compared with the measured discharge data (fig. 6). The goodness of the fit was evaluated by comparing two characteristic parameters – the time and the discharge of the wave peak. Manning's hydraulic roughness was used as the calibration parameter, separately for the September

scenario and for the March scenario.

The water exfiltration into the channel banks was simulated with the Richards equation. The hydraulic characteristics and the saturated hydraulic conductivity of the stream banks were assumed to be the same as the measured soil matrix characteristics on the surrounding fields. The modelling procedure was identical to the methodology described in (Zumr and Císlerová, 2010).





### 3.3 Sediment regime

The total amount of sediment released during the summer experimental campaign was 41.6 kg, and during the winter experimental campaign the amount was 124.5 kg. Assuming regular initial distribution and uniform release of the sediment, this represents 0.10 kg m$^{-1}$ of the channel for the summer campaign and 0.29 kg m$^{-1}$ of the channel for the winter campaign.

The maximum suspended solids concentration in profile A, with a value of 9 g l$^{-1}$, was observed at the moment when W1 and W4 were approaching (fig. 7). The minimum peak concentration of 1.7 g l$^{-1}$ was reached for wave W3. In profile C, the peak sediment concentration reached 7.5 g l$^{-1}$ and dropped to 3.9 g l$^{-1}$ for waves W2 and 3.7 g l$^{-1}$ for wave W3. The total sediment mass that passed the catchment outlet was 18.2 kg during the first wave, 13.8 kg during W2, and 9.6 kg during W3.

The amount of carried sediment measured at site C decreased from 48.5 kg to 30.7 kg between waves W4 and W6. The peak
concentration of suspended solids in profile C reached close to 8 g l$^{-1}$ for W5 and 5 g l$^{-1}$ for W6.

### 4 Discussion

#### 4.1 A comparison with the results from a natural catchment

The setup of our experiment was based on a study made by (Eder et al., 2014), who carried out two flushing experiments in a natural stream in the HOAL experimental catchment, Austria (Blöschl et al., 2016). The catchment is similar in size, climate,
soils and management to the Nučice catchment. The HOAL stream meanders through a forested belt. The monitored length is 590 m, with an average slope of 2.4 %. The stream cross-section is irregular, and the channel width varies from 0.6 m to 1.0 m. The longitudinal slope is relatively homogeneous over the whole monitored length, with the exception of significantly steeper parts around stationing 500 m (the initial part of the monitored section). Our experimental section had an inverse course, with the slope gradually increasing from 2.3 % in the first section of ca 70 m to 3.3 % in the last section ca 200 m in
length (nearly half of the length of the total monitored course).

The HOAL experiments were carried out in August 2011 on two days separated from each other by a gap of approximately one week. The volume of pumped water was 17 m3, and the wave propagation was monitored on three observation sites. The two recurrent waves had a very similar character. The average celerity was 0.22 m s$^{-1}$, which corresponds very well with the celerity measured in our experiment in summer conditions (waves W1 – W3), and reached about 50 % celerity in comparison
with our experiment in winter conditions (waves W4 – W6). The amount of water recovered in the closing profile was 79 % and 75 % of the pumped volume, which is comparable with Nučice experiments W1 to W3. However, the wave transformation in HOAL was extreme, and the peak discharges were reduced from 57 l s$^{-1}$ to 8.7 l s$^{-1}$ and 7.9 l s$^{-1}$ at the outlet profile. This is due to lowering of the longitudinal profile, and probably also due to higher surface roughness of the natural stream channel. In the HOAL experiments, only 7.0 kg and 7.7 kg of suspended solids were recorded, which is only about 50 % of the amounts
of sediment for the Nučice catchment in summer conditions, and 15 % of the amounts of sediment for the Nučice catchment in winter conditions.Both the HOAL experiment and the Nučice experiment resulted in major stream bed sediment



mobilization during the rising limb of the hydrograph. A similar sediment regime in channels has also been observed during large flood events on other streams (Guan et al., 2015). A comparison of the measured hydrographs and the physical and geometrical characteristics of the stream channels shows that, for the HOAL experiment, there is an extremely high flood wave transformation, with relatively low retention. This is partly due to the decreasing longitudinal slope of the river bed, and partly

due to the higher surface roughness of the natural stream channel. It shows that the transport capacity of the generated waves was exceeded and the amount of transported sediment decreases along the monitored course. In the Nučice experiment, the flood wave transformation was considerably lower. However, we observed relatively significant water retention in the first experiment in summer conditions (see the section in the Discussion referring to mathematical hydraulic modelling). We conclude that the transport capacity of the flood wave was exceeded during the HOAL experiment, and detached sediment

from upper, steeper parts of the experimental course was redeposited downstream, as in most of the natural streams monitored by (Naden et al., 2016). By contrast, during our experiment in Nučice, the transport capacity of the flood waves was not reached, either in summer conditions or in winter conditions. The sediment concentrations and also the fluxes therefore increased continuously throughout the section.

## 4.2 Stream potential for sediment trapping

We have to keep in mind that the artificial flood waves used in this experiment were relatively small in volume and of short duration. Based on the monitoring of the natural runoff events, we estimate that the minimum time needed for complete bedload sediment removal with comparable discharge is in the range of 10 to 24 hours (Zumr et al., 2015). Although the amount of sediment transported by the waves decreased within each set of experiments, there was still enough sediment left in the channel that may be released if there is a larger wave. The clockwise hysteresis of the sediment concentration – discharge relation

suggests that the sediment originates from nearby. Similar results were observed e.g. by (Molder et al., 2015; Seeger et al., 2004) The amount of resuspended sediment was significantly lower in summer conditions. We relate this to the particular conditions in the channel with dense erect vegetation and dry conditions, which led to storage of a considerable proportion of the water.

Significant changes in surface roughness, which also affect this process, may be documented by mathematical modelling of

25 the movement of the wave through the experimental section using a 1D hydraulic model, as will be discussed below.

These processes, though with reverse trends exhibiting a decrease in the amount of resuspended sediment over sections downstream, due to very different stream channel characteristics, have also been confirmed by the similar HOAL experiment, performed by (Eder et al., 2014). The results clearly show the potential of even well-trained channels without visible signs of sediment accumulation to release sediment during flood events.

The experiments showed that a well-trained stream can act both as a trap and as a sediment source. However, the hydraulic characteristics of the flood wave and the physical and geometrical characteristics of the channel will be crucial for indicating whether deposition or remobilization will occur in a given section and in a given event.



### 4.3 Temporal variability of sediment resuspension

All the sedigraphs show similar behavior (Fig. 5). The sediment concentration increases rapidly immediately after the arrival of the wave. The highest sediment concentration is always directly measured at the wave front, and does not necessarily correspond to the peak discharge. After culmination of the wave, the concentration of the sediment also decreases. The highest sediment concentration peak was observed when waves W1 and W4 were approaching, i.e. in the initial experiment of each campaign. We relate this to the fine-textured sediment that had been deposited in the stream during previous events. Our initial assumption was that most of the solid particles move only a short distance, because of low water velocity and short wave duration. Only the finest particles would be mobile enough to travel longer distances. However, our experiment showed that while the discharges decreased along experimental sections A to C, the sediment concentration increased (see Table 2, Table 3 and Fig.4). This suggests that the transport capacity of the stream had not been reached, even for lower discharges at the outlet point, and at least fine-textured soil particles were resuspended and transported over the whole observed section of the stream channel. To test this assumption, we estimated the maximum clear water transport capacity during the observed flow according to the simple transport capacity equation proposed by (Govers, 1990). The transport capacity of the peak flow was 990 gl$^{-1}$, which greatly exceeds the measured values. The sediment concentration behavior during waves W2 to W6 was similar (Fig. 5).

The remobilized sediment mass was two to three times higher in March than in September. We relate this to higher water velocity, as a result of which heavier particles contribute to the recorded amount, due to the higher transport capacity. It is not technically possible to measure the total initial mass of the sediment in the stream, and we can only make an estimate on the basis of previous runoff events that the conditions in September and March were similar, and were close to a quasi-steady state for the stream. In both cases, the last antecedent erosion event had taken place more than two months before the experiment, followed by at least one runoff event when no soil erosion was recorded and the discharge was above 5 l s$^{-1}$.

The experiments confirmed our assumption that vegetation development is a crucial parameter that affects flood wave retention and propagation, and also the sediment dynamics. Reverse vegetation conditions are documented by Fig.3 – fully erect well-developed dense vegetation in the summer set of experiments (W1 – W3) vs. no erect vegetation in the winter set of experiments (W4 – W6).

The general behaviour of the sediment transport during both sets of experiments (summer conditions vs. winter conditions) is the same, since the bedload sediment is available throughout the year. In both cases, it decreases event by event, but the sediment load increases along the sections. The general difference between the resuspension in fully-developed vegetation (summer conditions) vs. the winter conditions is 2.7 times higher for the first event and 3.2 times higher in winter time for the second and third events, as regards total transported sediment. As regards sediment concentrations, the peak values were the same for the first events and ca 50 % for the second and third events. Well-developed vegetation therefore significantly increased the trapping capacity of the stream channel (Keesstra et al., 2012).





The flood waves propagate differently in September and in March. While in September the successive waves speed up, in March the wave velocities are very similar for all experiments. When we compare the speed of flood propagation in a vegetated channel and in an empty channel, the March waves propagate twice as fast, and reach a 30 % higher peak discharge. The reason is twofold: (i) there is higher vegetation resistance in September, and (ii) there is higher baseflow and therefore a greater difference between the water velocity and the wave celerity in March. The volume of water recovered on site C is slightly larger during the March experiments. It should be noted that in reality the general summer and winter regime may vary because of variable rainfall patterns and catchment conditions (Buendia et al., 2016; Walling and Amos, 1999).

There is a significant water loss in the case of wave W1, which was released into an almost empty channel with dry stream banks (Table 1). Because of water exfiltration, interception on the vegetation, and filling of the streambed depressions along the 424 m long channel, the water loss reached 31 %. During all other experiments, including W4, the water loss was only 10 to 15 %.

A comparison of the simulated hydrographs and the measured hydrographs is presented in figure 6. It can be stated that the numerical simulations mimic the monitored hydrographs, and the effect of the vegetation seems to be a correct assumption. In the simulation of the September experiments, very high Manning roughness values (n = 0.20) were reached, while in March the optimized Manning n was equal to 0.12. The values fit well into the common ranges for sparsely-vegetated and densely-vegetated streams published e.g. by (Luhar et al., 2008) and by (Vereecken et al., 2006).

## 5 Conclusion

Our paper has presented the methodology for an artificial flood experiment conducted on an experimental agricultural catchment, and the results of the experiment. Three successive flood waves, each with an approximate volume of 16 m$^3$, were released into the upper part of the drainage channel. The aim was to monitor the transformation of the flood wave and the sediment transport within the channel.

On the basis of our results, we concluded that even well-trained and straight channels trap sediment, which can be mobilized by subsequent small floods.

The resuspension regime depends on the current conditions of the stream and the instream vegetation, and therefore changes significantly in the course of a year. The sediment moves quickly in spring and early summer, but in the later part of the year the channel serves as a sediment trap and the resuspension is slower.

The resuspension regime and the sediment loads within the succeeding small flood waves do not change considerably. The artificial waves that we initiated do not have sufficient magnitude to flush the bedload sediment out from the entire channel.





## Author contribution

David Zumr was manging the experiments, analysed data and prepared the manuscript with contribution of all the other co-authors. Tomáš Dostál and Jan Devátý co-organized the experiments. Petr Valenta did the HEC-RAS simulation. Pavel Rosendorf was responsible all the laboratory analyses. Peter Strauss and Alexander Eder participated on the experiments and the data evaluation.

## Competing interests

The authors declare that they have no conflict of interest.

## Acknowledgment

We thank our colleagues Josef Krása, Václav David, Petr Koudelka, Luděk Strouhal and Dan Fiala for their great help during the experiments. This research was prepared within the framework of Czech Science Foundation postdoctoral project GP13-20388P, Ministry of Agriculture projects NAZV QJ1230056, QJ1530181 and ÖAD WTZ Mobility project no. CZ18/2016 - 7AMB16AT002.

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





Table 1 Initial conditions before the experiments

| Waves no. | W1, W2, W3 | W4, W5, W6 |
|---|---|---|
| General conditions | Dry | Wet |
| Date | 4th September 2012 | 26th March 2013 |
| Baseflow ($l\ s^{-1}$) | 0.4 | 1.5 |
| Soil moisture conditions | Dry, bellow field capacity | Nearly saturated |
| Vegetation in channel | extensive | negligible |
| Air temperature (°C) | 12 | -2 |
| Baseflow water temperature (°C) | 7 | 2 |

Table 2 Hydrographs characteristics

| Wave no. | Profile | Time of first arrival (mm:ss) | Peak discharge ($l\ s^{-1}$) | Peak time (mm:ss) | Duration of limb rising | Duration of limb falling | Volume ($m^3$) | Wave celerity ($m\ s^{-1}$) | Wave velocity ($m\ s^{-1}$) |
|---|---|---|---|---|---|---|---|---|---|
| | A | 04:53 | 30.9 | 09:00 | 04:07 | 26:10 | 14.2 | 0.23 | n/a |
| W1 | B | 19:53 | 12.8 | 23:25 | 03:32 | 40:00 | 9.9 | 0.19 | n/a |
| | C | 34:35 | 12.3 | 40:35 | 06:00 | 40:00 | 9.8 | 0.22 | n/a |
| | A | 04:22 | 29.9 | 10:00 | 05:38 | 31:50 | 16.0 | 0.25 | n/a |
| W2 | B | 17:13 | 16.9 | 20:40 | 03:27 | 45:00 | 12.9 | 0.22 | 0.13 |
| | C | 31:29 | 17.7 | 34:29 | 03:00 | 47:00 | 13.7 | 0.22 | 0.15 |
| | A | 04:12 | 36.3 | 07:55 | 06:43 | 28:45 | 16.4 | 0.26 | n/a |
| W3 | B | 15:00 | 19.6 | 08:32 | 03:32 | 44:28 | 15.8 | 0.26 | n/a |
| | C | 29:40 | 19.6 | 33:40 | 04:00 | 44:00 | 14.8 | 0.22 | n/a |
| | A | 3:08 | 28.4 | 4:55 | 1:47 | 27:30 | 14.6 | 0.35 | n/a |
| W4 | B | 8:36 | 27.9 | 11:14 | 2:38 | 35:30 | 14.5 | 0.45 | n/a |
| | C | 16:28 | 22.8 | 19:23 | 2:55 | 41:00 | 13.3 | 0.43 | n/a |
| | A | 3:15 | 31.8 | 5:24 | 2:09 | 27:00 | 16.3 | 0.34 | 0.09 |
| W5 | B | 9:14 | 27.8 | 12:35 | 3:21 | 33:00 | 14.1 | 0.42 | 0.19 |





|      | C | 16:45 | 27.2 | 20:20 | 3:35 | 44:00 | 14.5 | 0.42 | 0.14 |
|------|---|-------|------|-------|------|-------|------|------|------|
|      | A | 2:51  | 37.2 | 8:06  | 5:15 | 28:30 | 16.3 | 0.39 | n/a  |
| W6   | B | 8:54  | 32.2 | 13:00 | 4:06 | 32:30 | 16.2 | 0.44 | n/a  |
|      | C | 15:58 | 29.6 | 18:33 | 2:35 | 44:00 | 14.1 | 0.44 | n/a  |

Table 3 Water and sediment budget as measured at the gauging stations (profile C)

| Wave no. | Inflow volume (m$^3$) | Outflow volume (m$^3$) | Cumulative sediment mass at the outlet (kg) |
|----------|-----------------------|------------------------|---------------------------------------------|
| W1 | 14.2 | 9.8 (69 %)  | 18.2 |
| W2 | 16.0 | 13.7 (86 %) | 13.8 |
| W3 | 16.4 | 14.8 (90 %) | 9.6  |
| W4 | 15.7 | 13.3 (85 %) | 48.5 |
| W5 | 17.2 | 14.5 (84 %) | 45.3 |
| W6 | 16.4 | 14.1 (86 %) | 30.7 |





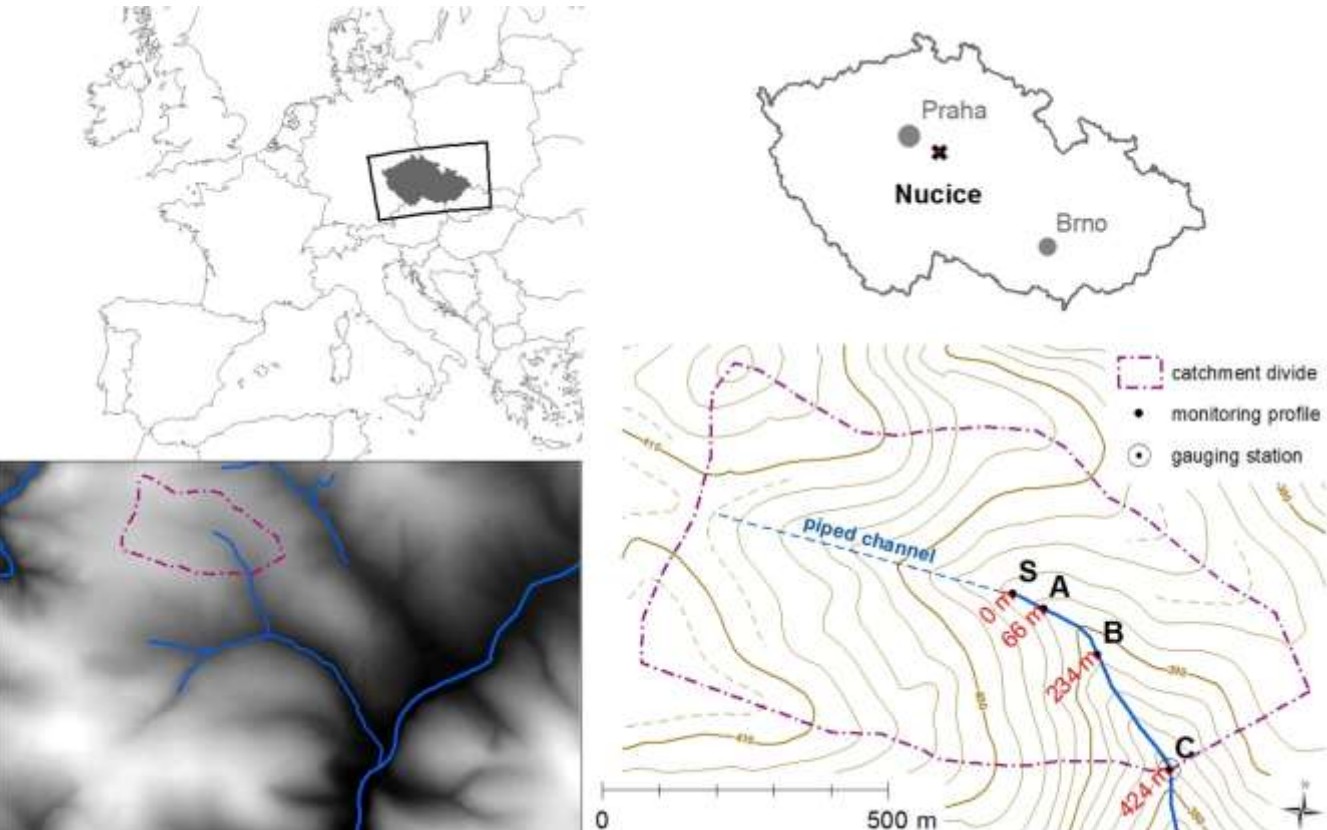

**Figure 1: Map of the Nučice catchment with the measurement sites.**





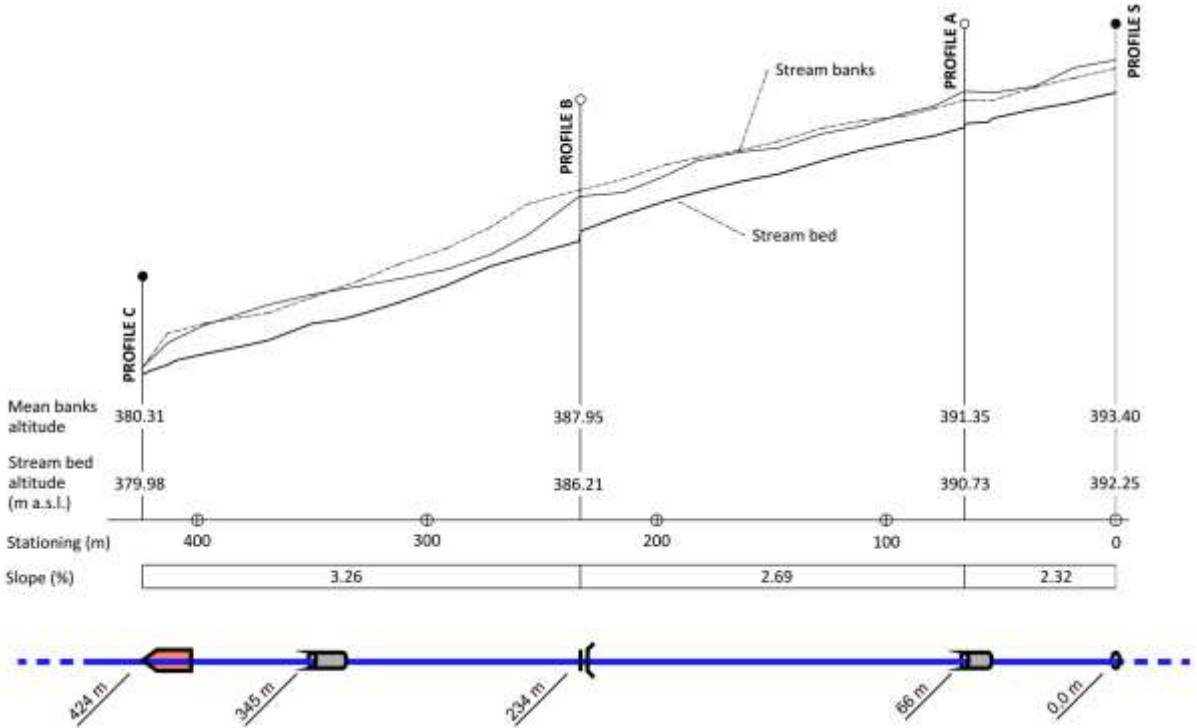

**Figure 2: Stream longitudinal profile with monitoring sites and culverts on the Nučice experimental catchment.**

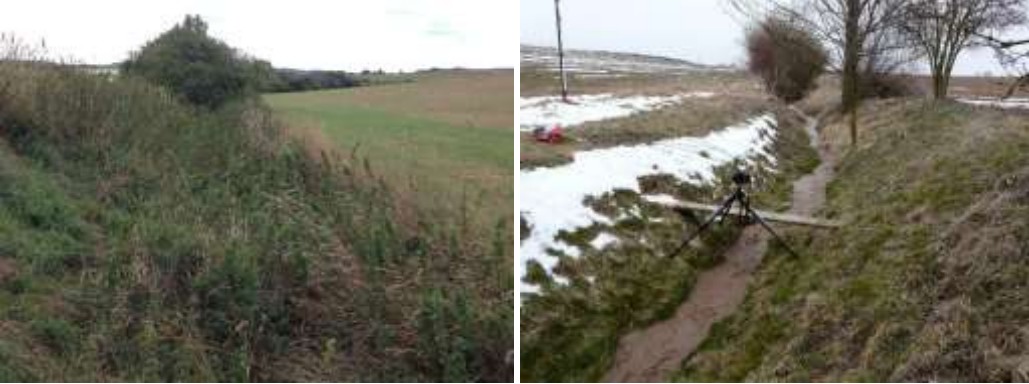

**Figure 3: Stream vegetation conditions during the two experiments. Dense instream vegetation in September 2012 (left) and no erect vegetation in March 2013 (right).**




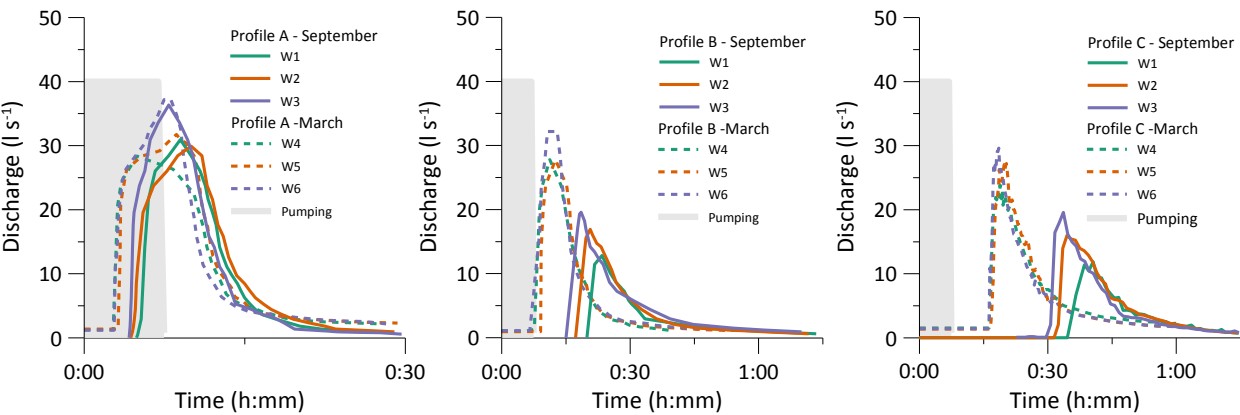

**Figure 4: Hydrographs of flood propagation along the monitored stream in the Nučice catchment. The different dynamics in September (W 1-3) and March (W 4-6) is caused by the current state of the stream and the vegetation conditions.**

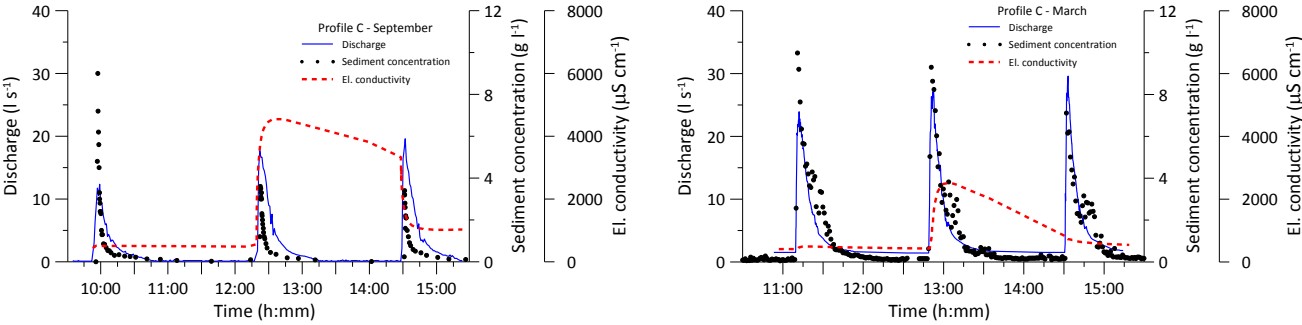

**Figure 5: Measured outflow rates and the concentration of suspended solids in the Nučice catchment outlet during experiments conducted in September 2012 (top) and in March 2013 (low). The sediment mobility in September was limited by the lower peak discharge rates due to high vegetation density and dry initial conditions.**





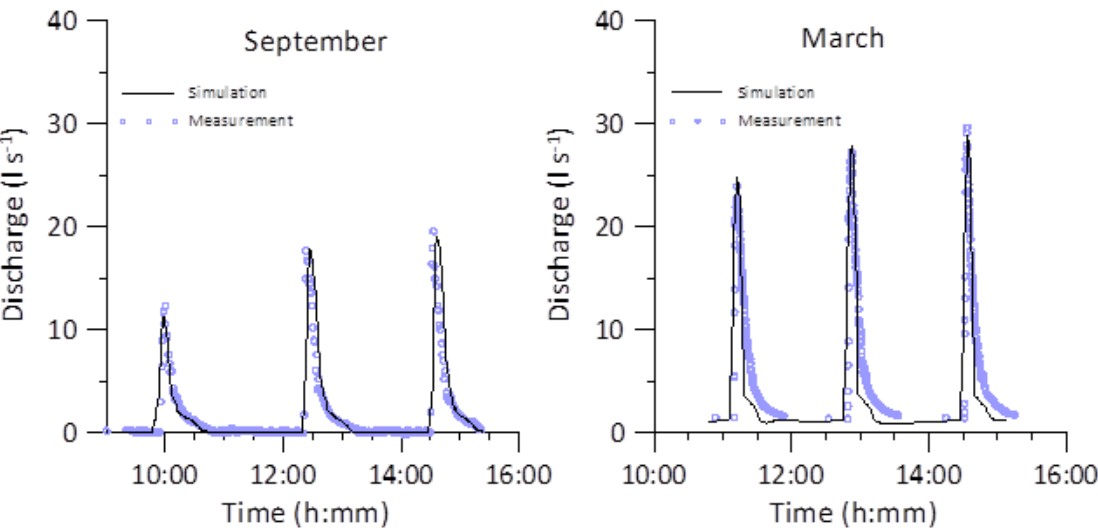

**Figure 6: Comparison of the flood wave characteristics measured at the gauging profile and simulated by HEC-RAS**

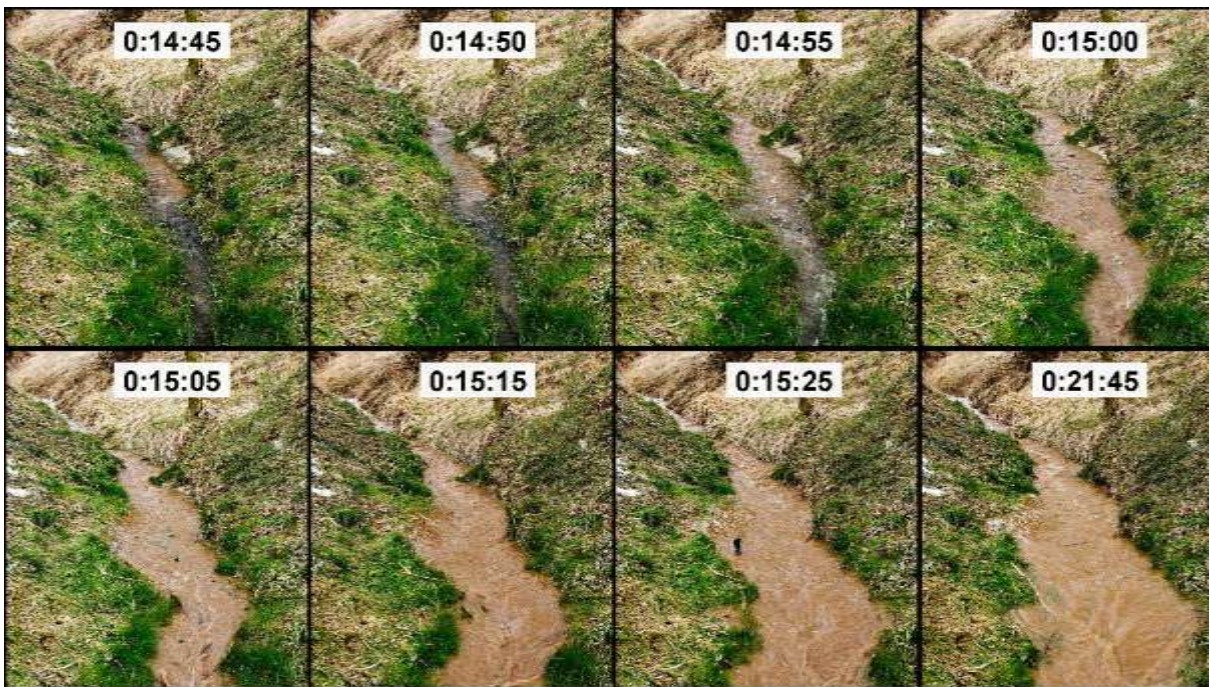

5    **Figure 7: The approach of the W4 wave front at stationing of 400 m. The times (h:mm:ss) stand for the duration from the start of the experiment.**