# Peer review of "Experimental determination of the flood wave transformation and the sediment resuspension in a small regulated stream in an agricultural catchment"

_Hydrology and Earth System Sciences, 2017_

## Referee Comment (RC1) · Anonymous Referee #1 · 5 Jul 2017

The paper presents a valuable methodology of conducting the artificial flood experiment in the field climatic conditions of Central Europe. Although the sets of experiments were conducted under varying conditions (considering the baseflow and vegetation cover) it presents values that could meet the interest of the scientific community.

To increase the scientific content and clarity of the text the following modifications are suggested:

1, Abstract should more clearly specify what was the aim of the study (as it was present

in the Conslusion part).

2, Regarding the content of the manuscript – part 3.2 Numerical modelling should be moved from the Result section to (Materials and) methods section. More detailed information on the setup should be provided on modelling in HEC-Ras (parametrization, steady flow?, etc.). Also I would consider joining Results and discussion (if the policy of the publisher allows it).

3, When citing the literature sources at the beginning or in the middle of the sentence, please use the format "author1 (year1), author2 et al. (year2) and authorC (year3)" – text highlighted in the draft of the manuscript. The format "(author1, year1)" is appropriate for the use at the end of sentence, as a reference for tables, etc.

4, Information regarding the period of experiments should be uniformed – in abstract (line 11 and 12), it is stated that the measurements were done in summer and spring, while elsewhere (e.g. page 3, line 2) summer and winter are referred to. The same applies to volume of the flood wave – 17 m3 (page 1, line 14) vs. 16 m3 (page 10, line 19).

5, Further I would suggest to refer to the experiments dates as September 2012 and March 2013 throughout the text (as it is done on page 5, line 3) and also tables and images (where appropriate).

6, The Latin names of the plant species should be written in italic script, except the "authority", e.g. Triticum aestivum L.

7, The script contains typos and formal errors (such as missing upper case, spaces, etc.) – some of them are also highlighted in the text.

8, Figures should be referred and Fig. 1 and 2 throughout the text, at this moment the format is not uniform.

9, The publisher does not require the number of issue of the cited journal articles, so these numbers should be removed.

10, In the final the manuscript should be checked by English native speaker to increase the readability of the text for the scientific community.

Please also note the supplement to this comment:
https://www.hydrol-earth-syst-sci-discuss.net/hess-2017-266/hess-2017-266-RC1-supplement.pdf

―――――――――――――――――

**Supplement:**

[revised manuscript text omitted]

---

## Referee Comment (RC2) · Anonymous Referee #2 · 7 Jul 2017

The paper presents an interesting experimental design at the Nučice agricultural catchment in the Czech Republic and represents a significant contribution in the fine sediment transport in constructed open channel drainages. The article is appropriate for the journal Hydrology and Earth System Sciences. The scientific methodology is sound, and methods explained thoroughly. The paper is well written and concise – a few editorial corrections are noted below.

Comments/Edits:

Title: The word "managed" would be a better word than "regulated".

Abstract is concise and well written.

Introduction: Page 2, line 3, the word "stacked" – not sure what that means – does it mean "embedded in the channel bed alluvium"?

Introduction: Page 2, lines 22-23, explain "more important" – this is a vague statement.

Introduction: Page 2, line 27: the word "here" can be replaced with "in this paper"

General formatting: ïČŸ Page 3, line 10, km2 ïČŸ Page 3, lines 15-16, 30-31: genesis, species names are italicized (check with journal); also correct throughout manuscript ïČŸ Page 3, line 19, artificially- trained ïČŸ Page 4, line 11, m3 ïČŸ Page 5, line 15, Figs. 4-6; line 19, Fig. 4; Page 6, line 4, Fig. 5; line 28, Fig. 6; Page 10, line 12, Figure 6. ïČŸ Page 7, line 23, m3 ïČŸ Page 7, line 31, a space is needed between "conditions. Both"

Experimental set-up: Page 4, lines 15 & 30: what is the size of the H flume?

Numerical Modelling: Page 6, line 9: Reword as: "The initial pumped water volume was 85% recovered in the C profile. . . . . .."

Numerical Modelling: Page 6, line 14: Best to state as :"simple 1D hydraulic model in HEC-RAS unsteady flow."

Numerical Modelling: Page 6, line18: Comment: Indirectly, stem blockage factor and frictional energy losses are fundamentally the same.

Numerical Modelling: Page 6, line 31-33: Is it possible to report with your use of the Richards equation, K, $\Phi$, and $\psi$ or h.

Discussion: Page 7, line 18, the word "convex" is better than "inverse"

Discussion: Page 8, lines 2-3, Comment: Was the HOAL experiment sediment-supply limited?

Discussion: Page 8, lines 30-33, Comment: It would be interesting to examine a long-term experiment observing a mass balance of fine sediment. I say that because your artificial water input was clear water (zero kg/s), but there was mass export. Just curious how that would change over time (hydrograph events) because it would potentially inform you better on shifts in source contributions over the annual seasons. Page 9, 2-4, was there any particle size distribution (PSD) data? That would also be interesting to observe over time. PSD requires an extensive commitment so I would not expect that that data are available.

Discussion: Page 9, line 23, the word "Reverse" – not sure what that means in the context of vegetation.

Discussion: Page 9, lines 26-28, Comment: Any discussion of the potential for fine sediment contributions from bank erosion?

Discussion: Page 10, lines 1-4, Comment: Others have found that soil moisture greatly affects erodibility of bank soils. You may want to reference this environmental conditions and interpolation of your findings.

---

## Author Comment (AC1) · 10 Aug 2017

We thank both reviewers for a generally positive feedback and the constructive comments and suggestions. We addressed and incorporated all the comments (incl. remarks in the supplement from reviewer #1) into the text of the manuscript. Our point by point responses to the comments follow.

Please, note the attached supplement with the updated manuscript, where all the revisions are highlighted.

[Figure]

@Referee #1

Comment 1, Abstract should more clearly specify what was the aim of the study (as it was present in the Conclusion part).

Response: We have added the information in the way as it is in the Conclusion (p.1, l.11).

Comment 2, Regarding the content of the manuscript – part 3.2 Numerical modelling should be moved from the Result section to (Materials and) methods section. More detailed information on the setup should be provided on modelling in HEC-Ras (parametrization, steady flow?, etc.). Also I would consider joining Results and discussion (if the policy of the publisher allows it).

Response: Part 3.2 was moved to Materials section (newly chapter 2.3). We have added more information on HEC-RAS modelling (section 2.3, paragraph on p. 5, l. 25). We decided not to merge the results and discussion. The journal recommends IMRaD sectioning, we feel that separation of results and discussion is beneficial in this case.

Comment 3, When citing the literature sources at the beginning or in the middle of the sentence, please use the format "author1 (year1), author2 et al. (year2) and authorC (year3)" – text highlighted in the draft of the manuscript. The format "(author1, year1)" is appropriate for the use at the end of sentence, as a reference for tables, etc.

Response: Corrected.

Comment 4, Information regarding the period of experiments should be uniformed – in abstract (line 11 and 12), it is stated that the measurements were done in summer and spring, while elsewhere (e.g. page 3, line 2) summer and winter are referred to. The same applies to volume of the flood wave – 17 m3 (page 1, line 14) vs. 16 m3 (page 10, line 19). Comment 5, Further I would suggest to refer to the experiments dates as September 2012 and March 2013 throughout the text (as it is done on page 5, line 3) and also tables and images (where appropriate).

Response to comments 4 and 5: The periods of the experiments are now referred only as September and March to prevent any confusion. The volume of the flood wave was corrected to 17 m3.

Comment 6, The Latin names of the plant species should be written in italic script, except the "authority", e.g. Triticum aestivum L.

Response: Corrected.

Comment 7, The script contains typos and formal errors (such as missing upper case, spaces, etc.) – some of them are also highlighted in the text.

Response: We have double-checked the text, the remarks highlighted in the reviewer's attached supplement were accepted and incorporated.

Responses to the comments in the reviewer's supplement: The exact information on the position of the conductivity meter during the 2012 and 2013 experiments was included (p. 5, l. 1) to explain why some data are missing in Table 2. Resolution of the final figures will be as requested by the HESS journal. The script size of the graphs will be unified. The actual non-uniformity was caused by a simple import of the pre-processed figures into MS Word template. We accept the suggested minor remarks to Fig. 1 and 2. We will correct the figures before the final submission.

Comment 8, Figures should be referred and Fig. 1 and 2 throughout the text, at this moment the format is not uniform.

Response: corrected

Comment 9, The publisher does not require the number of issue of the cited journal articles, so these numbers should be removed.

Response: Corrected, no. of issues was removed in the references.

Comment 10, In the final the manuscript should be checked by English native speaker to increase the readability of the text for the scientific community.

[Figure]

Response: We have double-checked the text again and corrected for typos and formal errors as well as nonuniformity of graphs and similar. We have submitted the text to a native speaker for proof reading.

@Referee #2

Comment 1, Title: The word "managed" would be a better word than "regulated".

Response: We suggest to keep the term regulated. The literature usually refers "regulated" as a river/stream with dams, weirs, embankments. The term "managed stream" is hard to find in similar papers.

Comment 2, Introduction: Page 2, line 3, the word "stacked" – not sure what that means – does it mean "embedded in the channel bed alluvium"?

Response: We replaced the term "stacked" with "embedded in the channel bed alluvium".

Comment 3, Introduction: Page 2, lines 22-23, explain "more important" – this is a vague statement.

Response: We reformulated the sentence to: "Similarly Huisman et al. (2013) showed that the previously suspended sediment is mobilized during the later parts of the year. In spring is the recently eroded sediment quickly flushed."

Comment 4, Introduction: Page 2, line 27: the word "here" can be replaced with "in this paper"

Response: The suggestion was accepted

Comment 5, General formatting: Page 3, line 10, km2, Page 3, lines 15-16, 30-31: genesis, species names are italicized (check with journal); also correct throughout manuscript, Page 3, line 19, artificially- trained, Page 4, line 11, m3, Page 5, line 15, Figs. 4-6; line 19, Fig. 4; Page 6, line 4, Fig. 5; line 28, Fig. 6; Page 10, line 12, Figure 6. , Page 7, line 23, m3, Page 7, line 31, a space is needed between"conditions. Both",

[Figure]

Experimental set-up: Page 4, lines 15 & 30: what is the size of the H flume? Numerical Modelling: Page 6, line 9: Reword as: "The initial pumped water volume was 85% recovered in the C profile......." Numerical Modelling: Page 6, line 14: Best to state as :"simple 1D hydraulic model in HEC-RAS unsteady flow.", Page 7, line 18, the word "convex" is better than "inverse"

Response: We corrected the formatting and typos, we rephrased the mentioned sentences as suggested. Size of the H flume was specified. : The H flume's inlet is 1.3 m wide and 0.68 m high, the straight approach length is 2 m.

Comment 6, Numerical Modelling: Page 6, line18: Comment: Indirectly, stem blockage factor and frictional energy losses are fundamentally the same.

Response: Yes, this is a very correct note, thank you for it. We have reformulated the sentence where we point out, that the considered vegetation effect is mainly due to friction around the stems, not due to water displacement by a huge amount of biomass in the stream (the two components of vegetation resistance as described by Green, 2005). The sentence now is: We assume that the vegetation resistance is mainly due to the stem blockage factor causing frictional energy losses, rather than by volume displacement effect, as the dense canopy occupies the cross section of the channel (Green, 2005).

Comment 7, Numerical Modelling: Page 6, line 31-33: Is it possible to report with your use of the Richards equation, K, $\Phi$, and $\psi$ or h.

Response: The van Genuchten's soil water retention parameters and saturated hydraulic conductivity were included: The water exfiltration into the channel banks was simulated with the Richards equation, van Genuchten's model for soil water retention curve was used. The hydraulic characteristics and the saturated hydraulic conductivity of the stream banks were assumed to be the same as the measured subsoil matrix characteristics on the surrounding fields (residual water content Thetar = 0.095, saturated water content ThetaS =0.44, Theta = 0.019 cm-1, n = 1.31, KS = 2.3 10-7 m
s-1).

Comment 8, Discussion: Page 8, lines 2-3, Comment: Was the HOAL experiment sediment-supply limited?

Response: Clearly not, in general we observe much higher sediment concentrations with higher flow rates during natural events which means that sediment concentration is not limiting.

Comment: 9, Discussion: Page 8, lines 30-33, Comment: It would be interesting to examine a long-term experiment observing a mass balance of fine sediment. I say that because your artificial water input was clear water (zero kg/s), but there was mass export. Just curious how that would change over time (hydrograph events) because it would potentially inform you better on shifts in source contributions over the annual seasons.

Response: We monitor the continuous sediment loads at the catchment outflow during the natural rainfall-runoff events, part of the data were published in Zumr et al. (2015). We were initially considering to compare the natural events with the artificial remobilization of the bedload sediment, but it is not trivial. The runoff events and natural sediment flux regime varies during the season – erosive rainstorms occur mostly in a short period in June and July, actual catchment conditions (such as initial topsoil saturation, topsoil compaction, crops) influence the regime as well. Generally, we assume that the sediment enters the stream and moves downstream during the spring snow melting. Another major sediment income is during early summer rainstorms, when the sediment is less mobile due to vegetated stream, part of it is trapped.

Comment 10, Page 9, 2-4, was there any particle size distribution (PSD) data? That would also be interesting to observe over time. PSD requires an extensive commitment so I would not expect that that data are available.

Response: We did not analyze the PSD of the mobilized sediment. This would have

needed much more water input (which we did not have) to obtain enough material for PSD data.

Comment 11, Discussion: Page 9, line 23, the word "Reverse" – not sure what that means in the context of vegetation.

Response: Word "reversed" was substituted with "contrasting".

Comment 12, Discussion: Page 9, lines 26-28, Comment: Any discussion of the potential for fine sediment contributions from bank erosion?

Response: The banks of the channel are in the most of the parts reinforced with concrete tiles or densely covered with grass (Fig. 3 and 7). We have not observed any bank erosion, not even during natural events when the discharge is by an order higher.

Comment 13, Discussion: Page 10, lines 1-4, Comment: Others have found that soil moisture greatly affects erodibility of bank soils. You may want to reference this environmental conditions and interpolation of your findings.

Response: We do not observe any sediment that originates from the stream bank erosion. In our case the banks water saturation may affect stream water exfiltration rate, but not the sediment input.

Please also note the supplement to this comment:
https://www.hydrol-earth-syst-sci-discuss.net/hess-2017-266/hess-2017-266-AC1-supplement.pdf

**Supplement:**

[revised manuscript text omitted]